# Combination of High Energy Intake and Intensive Rehabilitation Is Associated with the Most Favorable Functional Recovery in Acute Stroke Patients with Sarcopenia

**DOI:** 10.3390/nu14224740

**Published:** 2022-11-10

**Authors:** Yoichi Sato, Yoshihiro Yoshimura, Takafumi Abe, Fumihiko Nagano, Ayaka Matsumoto, Yoji Kokura, Ryo Momosaki

**Affiliations:** 1Department of Rehabilitation, Uonuma Kikan Hospital, Niigata 949-7302, Japan; 2Center for Sarcopenia and Malnutrition Research, Kumamoto Rehabilitation Hospital, Kumamoto 869-1106, Japan; 3Department of Nutritional Management, Keiju Hatogaoka Integrated Facility for Medical and Long-Term Care, Kanazawa 927-0023, Japan; 4Department of Rehabilitation Medicine, Mie University Graduate School of Medicine, Tsu 514-8507, Japan

**Keywords:** stroke, acute, energy intake, rehabilitation time

## Abstract

Energy intake and rehabilitation time individually contribute to the improvement of activities of daily living (ADL). This study aimed to investigate the additive effect of energy intake and rehabilitation time on ADL improvement in acute stroke patients with sarcopenia. The study included 140 patients (mean age 82.6 years, 67 men) with stroke. Energy intake during the first week of hospitalization was classified as “Sufficiency” or “Shortage” based on the reported cutoff value and rehabilitation time was classified as “Long” or “Short” based on the median. The study participants were categorized into four groups based on the combination of energy intake and rehabilitation time. The primary outcome was the gain of functional independence measure (FIM) motor during hospitalization. The secondary outcomes were length of stay and home discharge rates. Multivariate analysis was performed with primary/secondary outcomes as the dependent variable, and the effect of each group on the outcome was examined. Multivariate analysis showed that “long rehabilitation time and sufficient energy intake” (β = 0.391, *p* < 0.001) was independently associated with the gain of FIM motor items. The combination of high energy intake and sufficient rehabilitation time was associated with ADL improvement in acute stroke patients with sarcopenia.

## 1. Introduction

Sarcopenia is the aging-related loss of skeletal muscle mass and strength [1]. It leads to increased falls, functional dependence, and mortality [2]. Sarcopenia is a disease-specific feature. In patients with stroke, pathologies such as spasticity or flaccidity should be taken into account, and the upper extremities are more affected than the lower extremities [3].

Sarcopenia is associated with poor functional prognosis in patients with stroke. Sarcopenia before stroke onset is associated with poor gait function and even three months after stroke [4]. In the acute phase, stroke patients with low skeletal muscle mass have poor gait function at discharge [5]. In the convalescent phase, stroke patients with sarcopenia have poor activities of daily living (ADL) at discharge [6,7]. Sarcopenia is associated with poor outcomes in all phases of stroke. Therefore, prevention and treatment of sarcopenia are important for stroke patients.

Nutritional and exercise therapy is recommended to treat sarcopenia. In older patients with sarcopenia, nutritional therapy improves skeletal muscle mass and lower extremity muscle strength [8] and exercise therapy additionally improves gait function too [8,9]. Interventions combining nutritional and exercise therapy improve skeletal muscle mass, muscle strength, and physical function [8,9,10,11], as corroborated by a recent meta-analysis that reported the effects of nutritional and exercise therapy in older patients with sarcopenia [12]. Therefore, the combination of nutritional and exercise therapy might be important for the treatment of sarcopenia.

However, the evidence of nutritional therapy and rehabilitation in stroke patients with sarcopenia is unclear. Nutritional therapy for patients with stroke was reported to improve ADL by taking high-energy, high-protein supplements in addition to the usual hospital diet [13]. Standing exercises were reported to reduce the risk of sarcopenia in patients with stroke and contribute to improved ADL and shorter hospital stays [14]. Consequently, nutritional therapy and rehabilitation were shown to improve the functional prognosis of patients with stroke. These interventions were conducted by multidisciplinary teams of physicians, nurses, physical therapists, occupational therapists, speech therapists, dietitians, and pharmacists. Additionally, combined nutritional therapy and rehabilitation have been reported to improve ADL in patients with stroke in the convalescent phase [15,16]; however, there are few reports of the combined effect in the acute phase of stroke.

Therefore, this study aimed to investigate the effects of energy intake and rehabilitation time on ADL improvement in acute stroke patients with sarcopenia.

## 2. Materials and Methods

### 2.1. Participants and Setting

This single-center cross-sectional study was conducted at a 454-bed acute-care hospital in Japan between May 2020 and March 2022. The inclusion criteria for the participants were patients who were hospitalized within 48 h of the onset of cerebral infarction or cerebral hemorrhage. The presence of stroke was confirmed in all enrolled patients using computed tomography or magnetic resonance imaging and diagnosed by a medical doctor. Exclusion criteria of the participants were missing data, severely comatose (the Japan Coma Scale of three digits [17]), pacemaker insertion (from before admission to discharge), and death during hospitalization. After adopting the exclusion criteria, patients with sarcopenia were included in the analysis. Patients underwent thrombolytic therapy (tissue plasminogen activator), craniotomy decompression, or hematoma removal as needed. The observation period was from the date of admission to that of discharge from the acute-care hospital.

### 2.2. Diagnosis of Sarcopenia

Sarcopenia was diagnosed according to the consensus of the Asian Working Group for Sarcopenia 2019 [18] when both low muscle strength, as assessed by grip strength, and low skeletal muscle index (SMI), as assessed by bioelectrical impedance analysis (BIA), were present. The cutoff values of grip strength were less than 28 kg for men and 18 kg for women, and the cutoff values of SMI were less than 7.0 kg/m^2^ for men and 5.7 kg/m^2^ for women.

### 2.3. Rehabilitation during Hospitalization

Rehabilitation was provided by physical therapists, occupational therapists, and speech therapists. Regular conferences were held with physicians, nurses, physical therapists, occupational therapists, speech therapists, and dietitians to confirm the patient’s treatment plan. Rehabilitation programs (up to 3 h/day) were tailored to accommodate the individual patients’ functional abilities and disabilities, such as paralyzed-limb facilitation, range of motion exercises, basic movement training (mainly for the legs), walking training, resistance training, aerobic exercises using an ergometer aimed at improving endurance, ADL training, and dysphagia [19]. Japanese medical insurance allows a maximum of 3 h of rehabilitation per day after stroke, and it has been reported that sufficient rehabilitation from the acute phase is associated with a good functional prognosis after stroke [20]. Japanese guidelines for the management of stroke also recommend aggressive rehabilitation from the early stage of stroke onset [21].

### 2.4. Nutritional Management during Hospitalization

Nutritional management was also tailored to the patients’ functional and nutritional statuses [19]. Oral intake was recorded by the nurses based on how much the patient ate, using a scale of 0–10 and energy intake was calculated from the energy provided in food and the oral intake, which is generally adopted across Japanese hospitals [22,23]. Energy intake from enteral and intravenous nutrition was calculated from medical records and was added to the oral energy intake. Patients were provided high-energy and high-protein supplements as needed. Dietitians and a nutrition-support team monitored the patients’ nutritional conditions.

### 2.5. Data Collection

Patients’ information such as age, sex, body mass index (BMI), stroke type, stroke severity (based on the National Institute of Health Stroke Scale [NIHSS] score), comorbidities, primary treatment, duration of hospital stay, biochemical tests (serum albumin and hemoglobin), and discharge destination were collected from medical records. The NIHSS is a scale that assesses the neurological severity of stroke, including consciousness, speech, neglect, visual field loss, eye movement, ataxia, motor, sensory, and dysarthria [24]. Scores range from 0 to 42, with higher scores indicating more severity. The Geriatric Nutritional Risk Index (GNRI) was used as an index of nutritional status [25] which was calculated using the formula: [14.89 × serum albumin level (g/dL)] + [41.7 × (current body weight/ideal body weight)]. Ideal body weight was defined as a BMI of 22.0 kg/m^2^ [26], rather than calculated using the Lorentz formula, as there was no reported difference in GNRI when ideal body weight was calculated using the Lorenz formula or a BMI of 22.0 kg/m^2^ [26]. Grip strength was measured twice on the nonparalyzed side using a Smedley hand dynamometer (TTM, Tokyo, Japan), and the maximum value was considered (in the case of biparesis, the maximum value was measured on both sides). If the grip strength could not be measured due to impaired consciousness, such cases were excluded from the analysis. Appendicular skeletal muscle mass was assessed by a BIA device (Inbody S10, Tokyo, Japan). A physical therapist performed the assessments at least 3 h after eating. The SMI was calculated as appendicular skeletal muscle mass divided by the square of height. Higher SMI indicates more muscle mass. The Functional Oral Intake Scale (FOIS) [27] was used to assess swallowing function and was evaluated by the speech therapist. All assessments were performed within 5 days of admission, considering the initial acute treatment. Energy intake during the first week of hospitalization was collected from the medical records, and the average daily values were calculated [19]. Energy intake was calculated per actual body weight (ABW). Average rehabilitation time per day was calculated from the total time of physical therapy, occupational therapy, and speech therapy divided by the number of days in the hospital.

### 2.6. Outcomes

The primary outcome was the Functional Independence Measure (FIM) [28] motor item gain (FIM-motor gain). The FIM consists of 13 motor items and five cognitive items, rated on a scale from 1–7 points (complete dependence to complete independence). The FIM-total score ranges from 18 to 126, and the FIM-motor score ranges from 13 to 91. Higher scores indicated higher independence in ADL. The FIM-motor gain was defined as the difference between the FIM-motor item scores at discharge and admission. The secondary outcomes measured were length of hospital stay and home discharge rates.

### 2.7. Sample Size Calculation

In our previous study, the standard deviation (SD) of the FIM-motor gain in patients with acute stroke was 21 [19]. If the true difference between the groups was 20, it was estimated that at least 18 patients in each group would be needed to reject the null hypothesis with a statistical power of 0.8 and an error of 0.05 [29].

### 2.8. Statistical Analyses

Continuous data were presented as mean (SD) or median [25–75% percentile]. Categorical data were presented as the number (%) of individuals. 

Energy intake was defined as “Sufficiency” or “Shortage” based on a cutoff value of 20.7 kcal/ABW/day from a previous study on patients with acute stroke [19]. The rehabilitation time was defined as “Long” or “Short” based on the median rehabilitation time. Based on these categories, patients were divided into four groups: “Short × Shortage”, “Long × Shortage”, “Short × Sufficiency”, and “Long × Sufficiency”. Comparisons among the four groups were performed using a one-way analysis of variance and Tukey’s post hoc test as a post-test.

Multivariate analyses were used to examine whether the “Short × Shortage”, “Long × Shortage”, “Short × Sufficiency”, and “Long × Sufficiency” classifications were independently associated with FIM-motor gain, duration of hospital stay, and home discharge rate. Based on previous studies, age, sex, BMI, NIHSS, GNRI, SMI at admission, grip strength at admission, FIM-total at admission, and FOIS at admission were the covariates used to analyze the primary/secondary outcomes [19,30,31]. The variance inflation factor < 3 was defined as no multicollinearity [32]. All statistical tests were performed using SPSS version 28.0 software (IBM Corp., Armonk, NY, USA). Differences were considered statistically significant at *p* < 0.05.

## 3. Results

Figure 1 shows the study flowchart. The study initially included 485 patients with stroke. However, 68 patients were excluded due to missing data (*n* = 24), severely comatose (*n* = 18), pacemaker insertion (*n* = 12), or death (*n* = 14). Stroke patients without sarcopenia (*n* = 277) were also excluded. Finally, 140 patients were included in the study. 

Table 1 shows the basic demographic and clinical characteristics of the patients. The mean age of the cohort was 82.6 ± 9.6 years and included 67 men (47.9%). Patients in this study began rehabilitation within a median of one day from onset. The mean energy intake was 20.2 ± 7.9 kcal/ABW/day and the median rehabilitation time was 54.2 [41.2–69.8] min/day.

Table 2 shows the comparison of basic characteristics and home discharge rates among the subgroups. Age, sex, BMI, SMI, grip strength, and FIM did not significantly differ between the groups. The home discharge rate was also not significantly different. 

We examined the impact of energy intake and rehabilitation time on FIM-motor gain, as the primary outcome. Figure 2 shows the comparison of FIM-motor gain among the four groups. The FIM-motor gain score was significantly higher in “Long × Sufficiency” group than others (“Short × Shortage”: *p* < 0.001, “Long × Shortage”: *p* < 0.001, “Short × Sufficiency”: *p* < 0.001). Similarly, “Short × Sufficiency” group also showed a significantly higher score than “Short × Shortage”, (*p* = 0.009). The FIM-motor gain showed the best results when both energy intake and rehabilitation time were sufficient rather than insufficient respectively.

We examined the impact of energy intake and rehabilitation time on length of hospital stay, as the secondary outcome. Figure 3 shows the comparison of the length of the hospital stay. There were no significant differences among the groups (*p* = 0.104). 

We conducted multivariate analysis to adjust the influence of covariates for outcomes. Table 3 shows the results of the multivariate analysis for FIM-motor gain and length of hospital stay. “Short × Shortage” (β = −0.225, *p* = 0.009) and “Long × Sufficiency” (β = 0.391, *p* < 0.001) were independently associated with FIM-motor gain. In contrast, there were no significant associations with length of hospital stay for any variables. No multicollinearity was observed between the variables. After adjusting for covariates, FIM-motor gain was affected by both sufficiency and insufficiency of energy intake and rehabilitation time. Neither energy intake nor rehabilitation time was associated with length of hospital stay.

We conducted logistic regression to adjust the influence of covariates for home discharge rates. Table 4 shows the multivariate analysis of home discharge rates. There were no significant associations with home discharge rates for any variables. Neither energy intake nor rehabilitation time was associated with home discharge rates.

## 4. Discussion

The present study investigated the association between energy intake and rehabilitation time on ADL improvement in acute stroke patients with sarcopenia. The study findings imply the following: First, sufficient energy intake and longer rehabilitation time were associated with better ADL. Second, energy intake and rehabilitation time were not associated with the length of hospital stay and home discharge rates. The combination of sufficient energy intake and longer rehabilitation time was most strongly associated with ADL improvement in acute stroke patients with sarcopenia.

Sufficient energy intake and longer rehabilitation time improved ADL. Nutritional therapy and rehabilitation for patients with stroke are associated with good functional prognosis [29,33,34,35]. The use of high-energy (2 kcal/mL), high-protein (9 g/100 mL) oral nutritional supplements (ONS) for patients with stroke improves FIM-motor and 6-min walking distance compared to standard ONS (1 kcal/mL, 4 g/100 mL) [13]. High-frequency rehabilitation of patients with acute stroke i.e., at least twice/day, improves walking ability at discharge compared to low-frequency rehabilitation [20]. However, interventional strategies of combined nutritional therapy and rehabilitation were previously reported in the convalescent phase patients [14], but not in the acute phase. The findings of this study suggest that combining nutritional therapy and rehabilitation is associated with a good functional prognosis in patients with acute stroke. Hence, it is essential to provide both nutritional therapy and rehabilitation for faster recovery.

Energy intake and rehabilitation time were not associated with the length of hospital stay and home discharge rates. Malnutrition and sarcopenia lead to lower home discharge rates through impaired ADL [6,36]. Therefore, we predicted that functional improvement through aggressive nutrition and rehabilitation would have a favorable impact on the length of hospital stay and home discharge rates. However, our study did not show an association between energy intake and rehabilitation with the length of hospital stay or home discharge rates. In addition to physical function, various factors impact these outcomes, such as family presence, long-term care insurance, and housing environment [37,38,39], which were not studied. Similarly, this study did not consider factors that determine nutritional intake and rehabilitation time, such as appetite, motivation for rehabilitation, time of examinations, and the presence or absence of intravenous infusion. Due to the retrospective nature of the study, some unadjusted factors might have affected the outcomes. In addition, because length of hospital stay and home discharge rates were secondary outcomes, the study design may not have been appropriate for examining their association with nutritional therapy and rehabilitation. In the future, high-quality prospective studies are required to examine the impact of nutritional therapy and rehabilitation on the length of hospital stay and home discharge rates, including social backgrounds such as the presence of family members for support and social welfare services.

Nutrition and exercise therapy is important for acute stroke patients with sarcopenia. In older patients with sarcopenia, the combination of nutrition and resistance exercise is effective in improving the symptoms [40]. In older people undergoing rehabilitation in a convalescent hospital setting, a combination of branched-chain amino acid (BCAA) supplementation and rehabilitation improved ADL and muscle mass [41]. Similarly, even in patients with stroke, a combination of BCAA supplementation and rehabilitation improved ADL and symptoms of sarcopenia [15]. This study shows that aggressive nutritional therapy and rehabilitation might be essential for improving functional prognosis in patients with stroke. In clinical practice guidelines for rehabilitation nutrition, aggressive nutritional therapy, and rehabilitation are recommended, taking into account the pathophysiology of the disease [42]. Contrastingly, longer rehabilitation with low nutritional intake may promote catabolism and sarcopenia. This study suggests the importance of considering rehabilitation from nutritional status and considering nutritional therapy from rehabilitation. Future studies should also consider nutrition and rehabilitation (ONS and resistance exercise). In acute stroke patients with sarcopenia, early and appropriate nutritional management, and rehabilitation, along with treatment for the disease, might improve favorable outcomes.

This study had a few limitations. First, it was conducted at a single acute-care hospital in Japan, which limits the generalizability of our findings. Second, because of the retrospective study design, the effect of confounding factors on outcomes could not be fully adjusted. Furthermore, this study did not show any causal relationship between nutrition and rehabilitation that lead to improvement of ADL. The study did not consider factors that define appetite and rehabilitation time. Third, this study examined energy intake and rehabilitation time but did not include ONS or the proportion of resistance exercise. Therefore, high-quality prospective interventional studies are needed in the future.

## 5. Conclusions

Sufficient energy intake and longer rehabilitation time are associated with ADL improvement in acute stroke patients with sarcopenia. In addition, it is suggested that a more detailed consideration of nutrition and rehabilitation will lead to optimized patient management.

## Figures and Tables

**Figure 1 nutrients-14-04740-f001:**
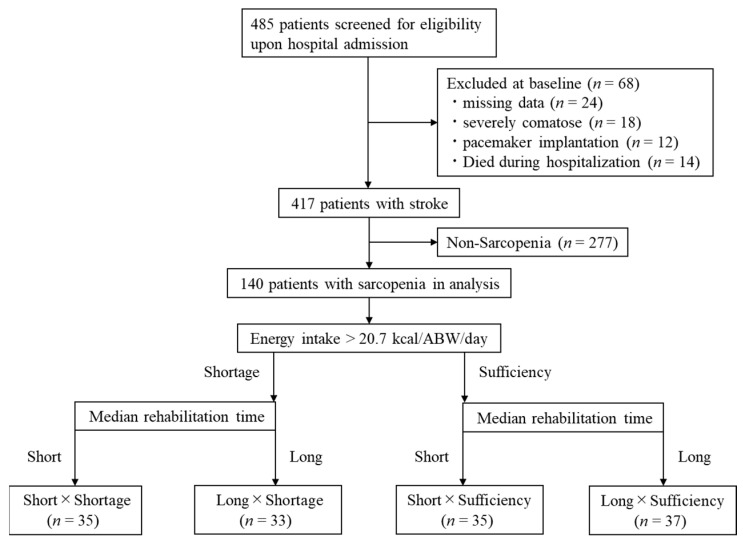
Flowchart of the study. ABW: Actual Body Weight.

**Figure 2 nutrients-14-04740-f002:**
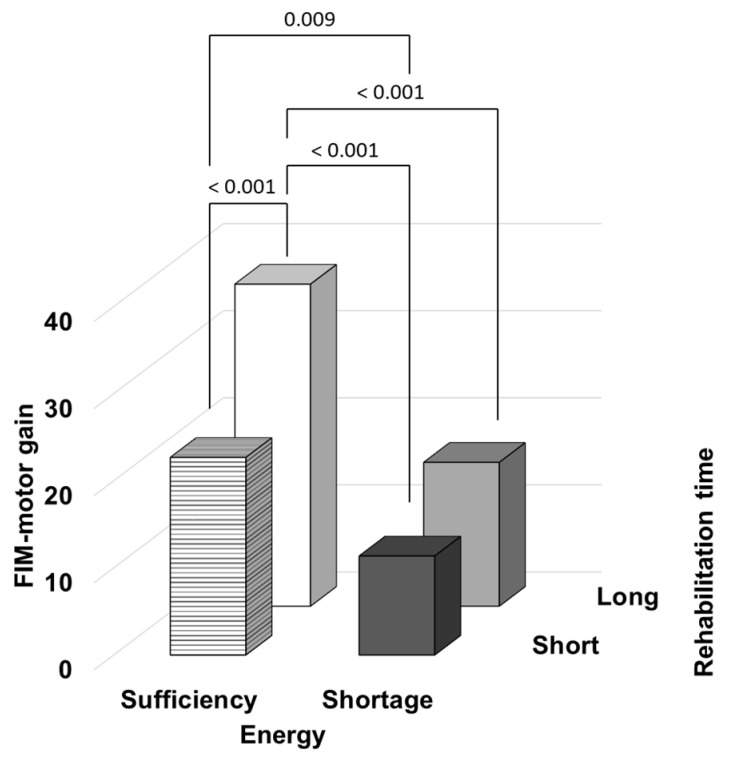
Comparison of the FIM-motor gain among the groups. The patients were categorized into four groups based on their energy intake during the first week of hospitalization (sufficient/shortage) and rehabilitation time (long/short). Black bar represents “Short × Shortage”, gray bar represents “Long × Shortage”, horizontally striped bar represents “Short × Sufficiency”, and white bar represents “Long × Sufficiency”.

**Figure 3 nutrients-14-04740-f003:**
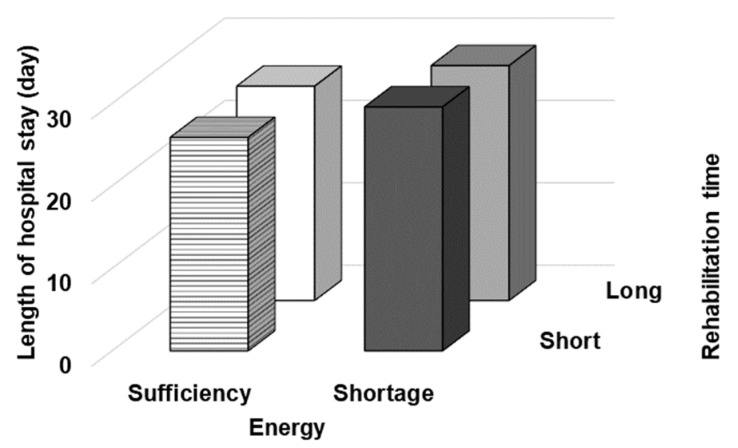
Comparison of length of hospital stay. The patients were divided into four categories based on their energy intake during the first week of hospitalization (sufficient/shortage) and rehabilitation time (long/short). Black bar represents “Short × Shortage”, gray bar represents “Long × Shortage”, horizontally striped bar represents “Short × Sufficiency”, and white bar represents “Long × Sufficiency”. There were no significant differences among the groups (*p* = 0.104).

**Table 1 nutrients-14-04740-t001:** Patient characteristics.

Variables	Total (*n* = 140)	Men (*n* = 67)	Women (*n* = 73)
Age (y)	82.6 (9.6)	78.6 (8.6)	86.4 (9.1)
Body mass index (kg/m^2^)	21.1 (3.6)	21.4 (3.9)	20.9 (3.2)
NIHSS score	6 [4–10]	5 [3–9]	7 [5–12]
Days from onset to rehabilitation (days)	1 [1–1]	1 [1–1]	1 [1–1]
Stroke type (infarct/hemorrhage)	107/33	52/15	55/18
Thrombolysis or Surgery (*n*)	7	3	4
Comorbidity (*n*)			
-Hypertension	91	40	51
-Diabetes	41	24	17
-Previous stroke	35	18	17
-Dyslipidemia	19	11	8
-Atrial fibrillation	41	20	21
Laboratory data			
-Albumin (g/dL)	3.9 (0.5)	3.9 (0.5)	3.8 (0.5)
-Hemoglobin (g/dL)	12.8 (1.9)	13.4 (2.0)	12.3 (1.6)
GNRI	98.1 (9.9)	99.8 (9.5)	96.5 (10.0)
Length of hospital stay (day)	27.5 (14.1)	27.9 (15.8)	26.2 (12.4)
FOIS at admission	4 [1–6]	5 [2–6]	4 [1–5]
Handgrip strength at admission (kg)	9.1 (8.2)	12.6 (9.1)	7.9 (5.6)
SMI at admission (kg/m^2^)	5.5 (1.0)	6.3 (0.5)	4.7 (0.7)
Energy intake during the first week of hospitalization (kcal/ABW/day)	20.2 (7.9)	20.3 (7.5)	20.1 (8.2)
-Oral intake	15.5 (10.8)	17.5 (10.1)	13.6 (11.2)
-Parenteral nutrition	3.0 (4.1)	1.9 (3.1)	4.1 (4.6)
-Enteral nutrition	1.7 (5.3)	0.9 (3.5)	2.5 (6.5)
FIM at admission			
-motor	30 [15–44]	33 [16–46]	29 [14–39]
-cognitive	17 [7–25]	16 [7–27]	18 [7–29]
-total	45 [24–67]	48 [25–75]	42 [20–66]
Rehabilitation time (min/day)	54.2 [41.2–69.8]	55.7 [42.9–75.4]	52.8 [40.3–64.2]
Means (SD) or median [25−75%tile]			
NIHSS: National Institute of Health Stroke Scale, GNRI: Geriatric Nutritional Risk Index, FOIS: Functional Oral Intake Scale,			
SMI: Skeletal Muscle Index, ABW: Actual Body Weight, FIM: Functional Independence Measure			

**Table 2 nutrients-14-04740-t002:** Comparison of the clinical characteristics among the groups.

	Short × Shortage (*n* = 35)	Long × Shortage (*n* = 33)	Short × Sufficiency (*n* = 35)	Long × Sufficiency (*n* = 37)	*p* Value
Age (y)	82.9 (10.4)	81.3 (11.0)	83.8 (6.4)	79.6 (9.1)	0.111
Men (*n*)	14	17	15	21	0.463
Body mass index (kg/m^2^)	21.3 (3.2)	22.0 (4.7)	20.7 (3.6)	20.5 (2.7)	0.308
NIHSS score	7 [6–10]	8 [5–13]	6 [4–9]	7 [4–10]	0.121
GNRI	97.1 (11.8)	102.1 (6.4)	95.5 (11.2)	97.8 (8.4)	0.092
FOIS at admission	4 [1,5]	5 [1,6]	5 [1,6]	5 [1,6]	0.332
Handgrip strength at admission (kg)					
-Men	12.6 (5.7)	10.8 (9.6)	13.5 (6.7)	12.6 (9.1)	0.099
-Women	7.5 (5.7)	7.4 (5.4)	7.9 (4.8)	8.2 (5.6)	0.105
SMI at admission (kg/m^2^)					
-Men	6.3 (0.5)	6.4 (0.4)	6.3 (0.4)	6.3 (0.5)	0.183
-Women	4.9 (0.4)	4.6 (0.6)	4.4 (0.8)	5.0 (0.4)	0.133
FIM at admission					
-motor	30 [15–44]	29 [15–40]	33 [16–45]	32 [15–46]	0.094
-cognitive	16 [7–27]	15 [7–24]	16 [7–29]	17 [8–26]	0.176
-total	45 [21–67]	43 [21–62]	45 [22–73]	47 [21–75]	0.123
Home discharge (*n*)	10	10	16	15	0.385
Mean (SD) or Median [25–75%tile]					
Short: Short rehabilitation time, Long: Long rehabilitation time, Shortage: Energy shortage, Sufficiency: Energy sufficiency					
NIHSS: National Institute of Health Stroke Scale, GNRI: Geriatric Nutritional Risk Index, FOIS: Functional Oral Intake Scale,					
SMI: Skeletal Muscle Index, FIM: Functional Independence Measure					

**Table 3 nutrients-14-04740-t003:** Multivariate regression analysis for FIM-motor gain and length of hospital stay.

	FIM-Motor Gain								Length of Hospital Stay							
	Model 1		Model 2		Model 3		Model 4		Model 1		Model 2		Model 3		Model 4	
	β	*p*	β	*p*	β	*p*	β	*p*	β	*p*	β	*p*	β	*p*	β	*p*
Short × Shortage	−0.225	0.009							−0.002	0.985						
Long × Shortage			−0.104	0.211							−0.002	0.981				
Short × Sufficiency					−0.073	0.388							−0.066	0.413		
Long × Sufficiency							0.391	<0.001							0.069	0.397
Short: Short rehabilitation time, Long: Long rehabilitation time, Shortage: Energy shortage, Sufficiency: Energy sufficiency																
β: Standardized partial regression coefficient Adjusted for age, sex, body mass index, National Institutes of Health Stroke Scale, Geriatric Nutritional Risk Index, Skeletal muscle mass index, handgrip strength, Functional Independence Measure-total, Functional Oral Intake Scale at admission																

**Table 4 nutrients-14-04740-t004:** Logistic regression analysis of home discharge rates.

	Model 1		Model 2		Model 3		Model 4	
	OR (95%CI)	*p*	OR (95%CI)	*p*	OR (95%CI)	*p*	OR (95%CI)	*p*
Short × Shortage	0.753 (0.186–3.056)	0.455						
Long × Shortage			0.511 (0.119–2.198)	0.367				
Short × Sufficiency					1.065 (0.920–1.232)	0.400		
Long × Sufficiency							0.620 (0.177–2.169)	0.455
Home = “1”, Other (rehabilitation hospital or nursing facility) = “0” Short: Short rehabilitation time, Long: Long rehabilitation time, Shortage: Energy shortage, Sufficiency: Energy sufficiency								
Adjusted for age, sex, body mass index, National Institutes of Health Stroke Scale, Geriatric Nutritional Risk Index,								
Skeletal muscle mass index, handgrip strength, Functional Independence Measure-total, Functional Oral Intake Scale at admission								

## Data Availability

The data are not publicly available owing to opt-out restrictions. Data sharing is not applicable.

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
