# Peer review of "Combination of High Energy Intake and Intensive Rehabilitation Is Associated with the Most Favorable Functional Recovery in Acute Stroke Patients with Sarcopenia"

_nutrients, 2022, doi:10.3390/nu14224740_

Round 1
Reviewer 1 Report
This study investigate the additive effect of energy intake and rehabilitation time on activities of daily living (ADL) improvement in acute stroke patients with sarcopenia at a single acute care hospital in Japan. They concluded that the combination of high energy intake and sufficient rehabilitation time is associated with ADL improvement in acute stroke patients with sarcopenia. I have some concerns before it can be officially accepted.
1. It is suggested to modify the title directly to the important conclusion of this paper.
2. Some abbreviated proper nouns must be described in more detail. For example, NIHSS, SMI, FIM, etc.
3. Line 17: The use of 'However' here looks inappropriate.
4. Results: The logical transition between different paragraphs is not clear, such as the sudden appearance of line 165 describing the difference in FIM-motor gain among the four groups. In the results section, some background introductions and descriptions of conclusions should be made.
5. Figure 2 & 3, Table 3-5, after describing the data, what conclusions can be drawn from these data must be elaborated. The author puts some conclusions in the discussion section.
Author Response
REVIEWER #1:
Comment 1: It is suggested to modify the title directly to the important conclusion of this paper.
(Response)
We agree with your comments. Following your advice, we changed the title as follows.
(Change)
(Title) “Combination of high energy intake and intensive rehabilitation is associated with the most favorable functional recovery in acute stroke patients with sarcopenia”
Comment 2: Some abbreviated proper nouns must be described in more detail. For example, NIHSS, SMI, FIM, etc.
(Response)
We agree with your comments. Following your advice, we added a description of the NIHSS, SMI, and FIM.
(Change)
(5th paragraph in the Materials and Methods section) “The NIHSS is a scale that assesses the neurological severity of stroke, including consciousness, speech, neglect, visual field loss, eye movement, ataxia, motor, sensory, and dysarthria [22]. Scores range from 0 to 42, with higher scores indicating more severity.”
(5th paragraph in the Materials and Methods section) “SMI was calculated as appendicular skeletal muscle mass divided by the square of height. Higher SMI indicates more muscle mass.”
(6th paragraph in the Materials and Methods section) “The FIM consists of 13 motor items and 5 cognitive items, rated on a scale from 1-7 points (complete dependence to complete independence). The FIM-total score ranges from 18 to 126, and the FIM-motor score ranges from 13 to 91. Higher scores indicated higher independence in ADL.”
Comment 3: Line 17: The use of 'However' here looks inappropriate.
(Response)
We agree. We removed the word "However".
(Change)
(Abstract) “This study aimed to investigate the additive effect of. . .”
Comment 4: Results: The logical transition between different paragraphs is not clear, such as the sudden appearance of line 165 describing the difference in FIM-motor gain among the four groups. In the results section, some background introductions and descriptions of conclusions should be made.
(Response)
We agree with your comments. We added a short introduction and conclusion to each paragraph within the Results section.
(Change)
(4th paragraph in the Results section) “We examined the impact of energy intake and rehabilitation time on FIM-motor gain, as the primary outcome. . . . FIM-motor gain showed the best results when both energy intake and rehabilitation time were sufficient rather than insufficient respectively.”
(5th paragraph in the Results section) “We examined the impact of energy intake and rehabilitation time on length of hospital stay, as the secondary outcome. . . .”
(6th paragraph in the Results section) “We conducted multivariate analysis to adjust the influence of covariates for outcomes. . . . After adjusting for covariates, FIM-motor gain was affected by both sufficiency and insufficiency of energy intake and rehabilitation time. Neither energy intake nor rehabilitation time was associated with length of hospital stay.”
(7th paragraph in the Results section) “We conducted logistic regression to adjust the influence of covariates for outcomes. . . . Neither energy intake nor rehabilitation time was associated with home discharge rates.”
Comment 5: Figure 2 & 3, Table 3-5, after describing the data, what conclusions can be drawn from these data must be elaborated. The author puts some conclusions in the discussion section.
(Response)
We agree with your comments. Short conclusions of figures and tables were added at the end of each paragraph in the Results section. Also, we added conclusions to the Discussion section.
(Change)
(4th paragraph in the Results section) “. . . . FIM-motor gain showed the best results when both energy intake and rehabilitation time were sufficient rather than insufficient respectively.”
(6th paragraph in the Results section) “. . . . After adjusting for covariates, FIM-motor gain was affected by both sufficiency and insufficiency of energy intake and rehabilitation time. Neither energy intake nor rehabilitation time was associated with length of hospital stay.”
(7th paragraph in the Results section) “. . . . Neither energy intake nor rehabilitation time was associated with home discharge rates.”
(1st paragraph in the Discussion section) “. . . . The combination of sufficient energy intake and longer rehabilitation time was most strongly associated with ADL improvement in acute stroke patients with sarcopenia.”

Reviewer 2 Report
Paper entitled: Combined effect of high energy intake and intensive rehabilitation on functional recovery in acute stroke patients with sarcopenia.
Overview: The aim of this study was investigate the Effects of energy intake and rehabilitation time on ADL improvement in acute stroke patients with sarcopenia.
General comments: This is a well-written and scientifically interesting study that addresses two importants areas of health, like nutrition and rehabilitation in stroke people. The introduction, I believe, does not fully justify the need for the study. The objective, is relevant, regarding the methodology I need more information,
Specific comments:
1. Introduction: There is talk of sarcopenia, nutrition... but you could add information about the team (multidisciplinary) that carries out the exercise therapy, the characteristics of the STROKE... There is a gradual loss of both skeletal muscle mass and strength with ageing (sarcopenia) that increases the risk of functional dependence, morbidity and mortality, and in stroke people this is more accussed. It is important to take into account spasticity and that patients usually have more sarcopenia in the upper limb than in the lower limb, and this information is not clare to me, in this manuscript…
- The keywords are absolutely fine.
- The methodology and results:
- we ask the authors the type of therapy for AVD in this patients with stroke.
- I have a question: Why do they work "up to 3h a day"??? Do you consider that this type of patients are capable of performing a therapy of more than 1h a day with good results?
- Tables 3, 4 and 5, if there were no significant associations for any variables, it is posible, to reduce to one or two tables?.
- In Line 170, describe Table 3, but it does not appear until line 184, the same happens with Table 4 and 5, described in lines 187 and 189 and appear in lines 202 and 206. Is it possible to modify this?. that is, write the description of each table just before the table itself and not a figure before.
4. All patients begin treatment within days of stroke? A patient after having suffered a stroke is essential to begin their rehabilitation treatment as soon as possible (whenever their condition allows it), in order to achieve functional objectives.
5. What kind of therapists have been working with these patients? Physiotherapist, occupational therapists, speech therapists... I consider it important to work multidisciplinary and to publicize the intervention of each professional, always for the benefit of the patient. I do not find it in the study.
6. Discussion and Conclusions… In line 213 and 226 “This study…” I suggest modified this.
7. The bibliography is current and encompasses the most recent scientific advances, while too many articles by the same author, who in turn is co-author of the manuscript, are being used. I would like to take into account studies to be highlighted on the subject such as (or another ones):
- McGlory C, van Vliet S, Stokes T, Mittendorfer B, Phillips SM. The impact of exercise and nutrition on the regulation of skeletal muscle mass. J Physiol. 2019 Mar;597(5):1251-1258. doi: 10.1113/JP275443. Epub 2018 Aug 1. PMID: 30010196; PMCID: PMC6395419.
- Li S, Gonzalez-Buonomo J, Ghuman J, Huang X, Malik A, Yozbatiran N, Magat E, Francisco GE, Wu H, Frontera WR. Aging after stroke: how to define post-stroke sarcopenia and what are its risk factors? Eur J Phys Rehabil Med. 2022 Oct;58(5):683-692. doi: 10.23736/S1973-9087.22.07514-1. Epub 2022 Sep 5. PMID: 36062331.
Author Response
REVIEWER #2:
Comment 1: Introduction: There is talk of sarcopenia, nutrition... but you could add information about the team (multidisciplinary) that carries out the exercise therapy, the characteristics of the STROKE... There is a gradual loss of both skeletal muscle mass and strength with ageing (sarcopenia) that increases the risk of functional dependence, morbidity and mortality, and in stroke people this is more accussed. It is important to take into account spasticity and that patients usually have more sarcopenia in the upper limb than in the lower limb, and this information is not clare to me, in this manuscript…
(Response)
Thank you for your suggestion. We agree with your comments. Following your advice, we added a paragraph to the introduction about sarcopenia and its characteristics in stroke. And we added about the team that conducts the rehabilitation.
(Change)
(1st paragraph in the Introduction section) “Sarcopenia is the aging-related loss of skeletal muscle mass and strength [1]. It leads to increased falls, functional dependence, and mortality [2]. Sarcopenia is a disease-specific feature. In patients with stroke, pathologies such as spasticity or flaccidity should be taken into account, and the upper extremities are more affected than the lower extremities [3].”
(4th paragraph in the Introduction section) “. . . These interventions are conducted by multidisciplinary teams of physicians, nurses, physical therapists, occupational therapists, speech therapists, dietitians, and pharmacists.”
Comment 2: we ask the authors the type of therapy for AVD in this patients with stroke.
(Response)
Thank you for your comments. Initial treatment for stroke that patients underwent in the acute phase included thrombolytic therapy, craniotomy decompression, and hematoma removal, but not stent therapy. We added treatment details to the Methods section.
(Change)
(“Participants and setting” paragraph in the Materials and Methods section) “Patients underwent thrombolytic therapy (tissue plasminogen activator), craniotomy decompression, or hematoma removal as needed.”
Comment 3: I have a question: Why do they work "up to 3h a day"??? Do you consider that this type of patients are capable of performing a therapy of more than 1h a day with good results?
(Response)
Thank you for your comments. Rehabilitation includes all physical therapy, occupational therapy, and speech therapy. Japanese medical insurance allows a maximum of 3 hours of rehabilitation per day after stroke, and it has been reported that sufficient rehabilitation from the acute phase is associated with a good functional prognosis after stroke (Ref.20). In addition, Japanese clinical guidelines for stroke treatment also recommend aggressive rehabilitation from the early stage of stroke onset (Ref.21). We added this information to the Methods section. If possible, aggressive rehabilitation was provided from early onset, depending on the patient's physical capacity and motivation.
(Change)
(“Rehabilitation during hospitalization” paragraph in the Materials and Methods section) “Japanese medical insurance allows a maximum of 3 hours of rehabilitation per day after stroke, and it has been reported that sufficient rehabilitation from the acute phase is associated with a good functional prognosis after stroke [20]. Japanese guidelines for the management of stroke also recommend aggressive rehabilitation from the early stage of stroke onset [21].”
Comment 4: Tables 3, 4 and 5, if there were no significant associations for any variables, it is posible, to reduce to one or two tables?.
(Response)
Thank you for your suggestion. We agree with you. Following your advice, we combined Tables 3 and 4.
(Change)
Tables 3 and 4 were combined and a new Table 3 was created.
(6th paragraph in the Result section) “Table 3 shows the results of the multivariate analysis for FIM-motor gain and length of hospital stay. "Short×Shortage" (β=-0.225, p=0.009) and "Long×Sufficiency" (β=0.391, p<0.001) were independently associated with FIM-motor gain. In contrast, there were no significant associations with length of hospital stay for any variables.”
Comment 5: In Line 170, describe Table 3, but it does not appear until line 184, the same happens with Table 4 and 5, described in lines 187 and 189 and appear in lines 202 and 206. Is it possible to modify this?. that is, write the description of each table just before the table itself and not a figure before.
(Response)
Thank you for your suggestion. We agree with you. Following your advice, we modified the layout of the tables and the text. The description of each table was moved to before the table.
Comment 6: All patients begin treatment within days of stroke? A patient after having suffered a stroke is essential to begin their rehabilitation treatment as soon as possible (whenever their condition allows it), in order to achieve functional objectives.
(Response)
We agree with your comments. Patients in this study began rehabilitation within a median of one day from onset. Therefore, the patients in this study started rehabilitation as early as possible. The number of days from onset to rehabilitation was added to Table 1 and the Results section.
(Change)
(2nd paragraph in the Results section) “Table 1 shows the basic demographic and clinical characteristics of the patients. The mean age of the cohort was 82.6 ± 9.6 years and included 67 men (47.9%). Patients in this study began rehabilitation within a median of one day from onset.”
Comment 7: What kind of therapists have been working with these patients? Physiotherapist, occupational therapists, speech therapists... I consider it important to work multidisciplinary and to publicize the intervention of each professional, always for the benefit of the patient. I do not find it in the study.
(Response)
We agree with your comments. Rehabilitation was mainly conducted by physical therapists, occupational therapists, and speech therapists. However, regular conference was held including physicians, nurses, and dietitians to discuss the treatment plan. Added to the Methods section.
(Change)
(“Rehabilitation during hospitalization” paragraph in the Materials and Methods section) “Rehabilitation was provided by physical therapists, occupational therapists, and speech therapists. Regular conferences were held with physicians, nurses, physical therapists, occupational therapists, speech therapists, and dietitians to confirm the patient's treatment plan. ”
Comment 8: Discussion and Conclusions… In line 213 and 226 “This study…” I suggest modified this.
(Response)
Thank you for your suggestion. We agree with your comments. Following your advice, the phrase "this study" was removed.
(Change)
(2nd paragraph in the Discussion section) “Sufficient energy intake and longer rehabilitation time improved ADL.”
(3rd paragraph in the Discussion section) “Energy intake and rehabilitation time were not associated with the length of hospital stay and home discharge rates.”
Comment 9: The bibliography is current and encompasses the most recent scientific advances, while too many articles by the same author, who in turn is co-author of the manuscript, are being used. I would like to take into account studies to be highlighted on the subject such as (or another ones):
(Response)
Thank you for your suggestion. We agree with your comments. Following your advice, we reduced the citations of co-authored papers (Ref.12) and added the papers you suggested (Ref.2, 3).
(Change)
(1st paragraph in the Introduction section) “It leads to increased falls, functional dependence, and mortality [2]. Sarcopenia is a disease-specific feature. In patients with stroke, pathologies such as spasticity or flaccidity should be taken into account, and the upper extremities are more affected than the lower extremities [3]. ”

Reviewer 3 Report
This single-center cross-sectional study was conducted in an acute care hospital in 148 sarcopenic patients admitted within 48 hours of the onset of cerebral infarction or cerebral hemorrhage.
In these patients, during the first week of hospitalization, the convergent and additive effects of energy intake classified as sufficient or poor and rehabilitation time ( long or short ) on two main goals were analyzed :
1- Improvement in activities of daily living (ADLs)
2- Variation in time of discharge to home.
Taking into account that each of the elements analyzed (energy intake, rehabilitation time) have a significant effect on the two goals, the scope of the project was to analyze through multivariate statistical comparison, the presence of any additive effects of the four experimental groups.
The results of the study indicate that:
1- sufficient energy intake and longer rehabilitation time were associated with improved ADLs and that a probable additive effect of the two parameters exists.
2- no additive effect of the two factors was observed on home discharge time.
In conclusion, this study confirmed what is already known from the literature, namely, that sufficient energy intake and longer rehabilitation time improve ADLs. Rather, the original finding concerns the additive effect of the two parameters.
The second objective that the authors had set for themselves, that concerning the patient's discharge time, which would certainly also have an economic significance in his or her hospitalization, did not yield significant findings.
The experimental design, from the choice of subjects recruited to the statistical analysis of the data is well structured.
The measurements taken both as an expression of muscle strength and the ability to perform movements (FIM), are correct.
The analysis of the energy component, although complex and operationally difficult, appears to be adequate.
The Figures and Tables are clear and make it possible to understand the results obtained even without reading the entire text completely.
For the reasons stated above, in recommending the publication of the manuscript, I endorse what the authors write in the conclusion:
"it is suggested that a more
detailed consideration of nutrition and rehabilitation will lead to optimized patient management".
Author Response
REVIEWER #3:
(Response)
We sincerely appreciate your thoughtful, positive and supportive comments. Your comments are very beneficial for our future research.

Round 2
Reviewer 1 Report
I agree accept in present form.
Reviewer 2 Report
I think the manuscript has improved with all the changes they have made. Thanks to the authors for their efforts.